# Spatio-temporal analysis of human brucellosis distribution in Neyshabur, Iran from 2015 to 2022, a cross-sectional study

Ayoub Fathabadi[1], Fatemeh Khorashadizadeh[2], Reza Darrudi[1], Mohammad Maroosi [3]*

1 Department of Health Information Technology, Faculty of Health and Paramedicine, Neyshabur University of Medical Sciences, Neyshabur, Iran, 2 Department of Epidemiology and Biostatistics, Neyshabur University of Medical Sciences, Neyshabur, Iran, 3 Department of Environmental Health Engineering, Faculty of Health and Paramedicine, Neyshabur University of Medical Sciences, Neyshabur, Iran

* maroosim@nums.ac.ir

## Abstract

### Background

Human brucellosis is a global endemic disease and a significant public health concern. This cross-sectional study aims to explore the characteristics of human brucellosis, as well as its spatial and temporal clustering.

### Methods

This cross-sectional study analyzed the population infected with human brucellosis in Neyshabur, Iran from 2015 to 2022. Data were obtained from the Neyshabur health center. Spatial analysis was conducted using Anselin Local Moran's I and Global Moran's I with ArcGIS 10.8.2. Descriptive statistical analyses were performed using SPSS 26.

### Results

During the study period, a total of 4568 brucellosis patients were reported. The incidence of disease was higher in men with 55.37%. The majority of patients belonged to the 30 to 39-year-old age group. Temporal clustering analysis revealed a peak incidence between March and June. The highest incidence rates were observed in spring and summer, with 34.76% and 33.84% of cases, respectively. Spatial clustering analysis indicated a higher prevalence in the southern and southeast districts of Neyshabur. The results of Global Moran's I analysis indicated that the distribution of brucellosis was not clustered in Neyshabur (P-value > 0.05).

### Conclusion

The findings of this study highlight the importance of educational programs for mothers and animal vaccination strategies, particularly in the months leading up to the peak brucellosis season in high-incidence areas. Further research is needed to assess the coverage of livestock vaccination and its impact on human brucellosis rates.

**Data availability statement:** The data used in this study was obtained from the Neyshabur Health Center as offline documents and Excel files. As the authors do not own the data, we are unable to provide a direct link or public access. Additionally, the Neyshabur Health Center is a local service-oriented institution that lacks the infrastructure for handling international correspondence, especially in languages other than Persian. There is no established protocol for responding to data requests from international researchers, which limits our ability to facilitate direct contact for data access. The data includes sensitive information such as geographical coordinates of patients, which imposes additional restrictions on sharing due to privacy and ethical considerations. Despite these constraints, the authors are committed to sharing the data with individual researchers upon request, provided that such requests comply with the ethical guidelines and constraints under which the data was collected. The data will be kept safely and securely in a personal and online encrypted cloud repository to ensure persistent or long-term data storage and availability. For inquiries regarding data access, please contact Mohammad Maroosi (maroosim@nums.ac.ir), Ayoub Fathabadi (ayofathabadi@gmail.com), Fatemeh Khorashadizadeh (fkhorashadi89@yahoo.com) and Reza Darrudi (darrudir1@nums.ac.ir). Access requests will be evaluated on a case-by-case basis in accordance with applicable ethical and legal standards, including those outlined in the IRB-approved protocol or patient consent form. Although the authors cannot make their study's data publicly available at the time of publication, all authors commit to make the data underlying the findings described in this study fully available without restriction to those who request the data, in compliance with the PLOS Data Availability policy. For data sets involving personally identifiable information or other sensitive data, data sharing is contingent on the data being handled appropriately by the data requester and in accordance with all applicable local requirements.

**Funding:** This study was supported by the Student Research Committee (SRC) funds of Neyshabur University of Medical Sciences (NUMS). The funders had no role in study design, data collection and analysis, decision to publish, or preparation of the manuscript.

**Competing interests:** The authors have declared that no competing interests exist.

## Introduction

Human Brucellosis is one of the seven neglected and under-detected acute febrile diseases that remains one of the major causes of morbidity throughout the world [1]. It results from the transmission of the brucellosis bacterial agent from infected animals to humans [2]. In populations with the traditional lifestyles, inhalation, direct contact with animal and consumption of unpasteurized dairy products and undercooked meat products are the primary modes transmissions leading to human brucellosis [3,4]. Human brucellosis is endemic in Iran were many people, especially those in rural areas, commonly keep livestock at their residence [5], This traditional lifestyle significantly contributes to the spread of brucellosis within the Iranian population [6]. The incidence of human brucellosis in Iran ranges from 7 to 276 per 100,000 population [7]. Based on previous research, Iran ranks fourth in the world in terms of incidence rate of human brucellosis [8–10]. Although the highest incidences rates of human brucellosis are related to the west and northwest regions of the Iran [7], the most frequent cases of human brucellosis pertain to Razavi Khorasan, the second most populous province in northeast of Iran [11]. According to Norouzinezhad, the incidence of human brucellosis was more than 20 per 100,000 population in Razavi Khorasan between 2009–2016 [12]. This relates to occupational, especially working with animals, ethnic and regional aspects of individuals' lifestyle. There is now a strong body of evidence to support that brucellosis can cause a reduction in quality of life [13], herd productivity, and food security [14] Considering human health, this disease affects the body's immune system leading to flu-like symptoms such as fever, weakness, malaise, weight loss and fatigue [15]. The medical, veterinary and socioeconomic implications are emphasized as the other significant public health concerns associated with brucellosis [15]. Brucellosis prevalence has been reported to be highest in some areas such as Central Asia, Mediterranean countries, Latin America, and sub-Saharan Africa. Due to the increasing trends in international traveling, trade and migration, the prevalence of the disease has increased in all ages of the population [16]. The confirmed cases are estimated in studies in various Asian countries with variable prevalence, including India (16.7%), Turkey (8.8%), Saudi Arabia (12.7%), Iraq (12.6%), and Iran (6.59%) [17]. However, a significant reduction in the incidence of cases has been observed in some countries, including Australia, New Zealand, and Japan [14]. This suggests that the prevention or even elimination of human brucellosis may be feasible which requires the prevention programs to achieve the desired outcomes. Iran, also, is endemic for human brucellosis, notably prevalent in areas where people reside in close contact with infected animals or animal products [10,18]. By 2017, the annual disease rate was estimated at one case per 100,000 population that the highest rate was reported in the western and northern regions of the country [7]. According to the Iranian Ministry of Health (MOH), Razavi Khorasan province, with an average of 21–30 per 100,000 population annually, is among the provinces with the highest incidence rate of human brucellosis in Iran [18]. Neyshabur, the second-largest city in the province, has over350 villages [19], where traditional livestock farming is common. This increases risk of infection for all members of family due to direct contact with contaminated domestic animals. Further, prevalence of brucellosis varies by time and place [20,21]. Between 2009 and 2016, the incidence of human brucellosis in Neyshabur ranged from 25.66 to 73.04 cases per 100,000 population [12]. increasing public awareness and improving intervention and preventive policies can reduce the incidence of brucellosis in high-risk areas [22]. With emergence of the new technologies in the field of health, Geographical Information Systems (GIS) have demonstrated the capability to discover the new epidemic pattern and hotspot areas to precisely provide policy guidance for brucellosis control [23] and significant reduction associated with the costs, care and prevention programs based on the geographical data [24]. Using the data from GIS in epidemiological investigations into the prevalence of brucellosis in Iran, Shirzadi et.al

(2021) identified the high-priority areas to implement the targeted interventions to control the disease [25]. According to Azizi et.al (2019), because of traditional consumption habits of unpasteurized fresh and local dairy products, Neyshabur stands out as a highly polluted and susceptible region to brucellosis, considering the pattern of livestock distribution, the spatial distribution of this disease varies in villages [26].

The incidence rate of brucellosis various by time and region, with different distributions of disease and incidence rates are reported across provinces in Iran [18,27–31]. In the present study, we employed a spatiotemporal distribution model to analyze local surveillance data on human brucellosis in Neyshabur, Razavi Khorasan Province, in the Northeast Iran, between 2015 and 2022. Additionally, we focused on investigating the characteristics of human and spatiotemporal distribution characteristics, as well as correlations within human brucellosis epidemics. This analysis aims to provide an overview of the human brucellosis situation in Neyshabur to inform future disease control decisions.

## Materials and methods

### Study area

This cross-sectional study was conducted in Neyshabur, the north-eastern region of Iran, with a population of 451,780 [32]. Neyshabur with a 5.653 $km^2$ area, located in 36° 25′ 26″ N, 58° 37′ 31″ E [33], is the second-largest city in the Khorasan Razavi province and the third-largest city in Eastern Iran. Neyshabur has 418 habitable villages organized into census blocks, which are the smallest spatial units in the cities of Iran. For this study, rural districts, a political division in Iran, were utilized as the geographical scale for conducting the spatial analyses.

### Data sources and patients

The data was collected from two sources comprising both non-spatial and spatial. Firstly, the data on human brucellosis cases was obtained from the Neyshabur health center University of Medical Sciences from 2015 to 2022. The criteria for diagnosing patients with human brucellosis were a positive Wright and 2-mercaptoethanol (2ME) tests in the population suspected of being infected. Secondly, the spatial divisions map of Neyshabur area was acquired through the municipal authorities. Because the lack of registered latitude and longitude for human brucellosis cases, we had to geocode patients' addresses using Google My Maps. To ensure patient privacy, we used the geomasked technique. Each patient's address was randomly assigned within a radius of half a kilometer around his/her home. Because of the study design that was executed at the district level, the probability pointing address of patients did not limited our study. All maps and spatial analyses in this study were created by the authors using ArcGIS software, version 10.8.2.

### Ethics approval and consent to participate

This retrospective study, approved by the Research Ethical Committee of Neyshabur University of Medical Sciences under the ethical code IR.NUMS.REC.1401.042, is in compliance with ethical standards. After the conformation of the study, the data related to human participants was obtained from the Neyshabur health center in a as anonymized format. This means that data were de-identified at the source by trained personnel. Additionally, the study team is committed to using these data only for the present study, and no one outside the study team has access to the data.

### Anselin local Moran's I

Anselin's Local Moran I is a local spatial autocorrelation measure derived from the global Moran's I statistic. It calculates the Moran's I value for each individual spatial unit (e.g., district) by creating a neighborhood around that unit and comparing its value to the overall

study area [34]. in this study, the spatial statistic was applied to identify the high and low incidence districts of Neyshabur considering human brucellosis incidence between 2015 to 2022. The output of Anselin's Local Moran I analysis is a new map that classifies the districts of Neyshabur into five classes based on the spatial patterns of human brucellosis incidence (High-High cluster, Low-Low cluster, High-Low cluster, Low-High cluster, not cluster).

## Global Moran's I

Global Moran's I is a spatial autocorrelation statistic that simultaneously measures spatial auto-correlation in a study area based on the attribute values of interest [34]. In this study, we used Global Moran's I to assess the distribution pattern of human brucellosis incidence in Neyshabur between 2015 and 2022. The null hypothesis states that the spatial distribution of human brucellosis incidence in each year is random. The interpretation of Global Moran's I is based on the P-value and Z-score. The null hypothesis may be rejected if the P-value is statistically significant ($P < 0.05$). If p-value is statistically significant, and the z-score is positive, it means that the high and/or low values of human brucellosis incidence in the dataset is more spatially clustered than would be expected. conversely, if the p-value is statistically significant, and the z-score is negative, it means that the spatial distribution of high values and low values of human brucellosis incidence in the dataset is more spatially dispersed than would be expected [35].

## Results

Between 2015 and 2022, a total of 4,568 cases of human brucellosis were reported in Neyshabur. The average incidence rate over these eight years was 108.26 cases per 100,000 population. The annual incidence rates indicate an increasing trend from 2018 to 2022, while no consistent trend was observed in other years. Throughout the study period, men were diagnosed with brucellosis, 55.36% more than women. Additionally, 89.32% of the patients were rural residents. The highest incidence rates were observed in the summer and spring, with 34.76% and 33.84% of cases, respectively. See Table 1 for more details.

The median age of patients is 38 years, with an interquartile range of 25 to 52 years. A significant relationship between age and gender was observed ($P < 0.001$). See Fig 1.

Monthly incidence data for brucellosis reveal that the lowest rates of infection occur in December, while the highest rates are observed in June. An increasing trend in incidence is noted from December to June, followed by a decreasing trend from June to December. See Fig 2.

Table 1. The characteristics of Seasonally incidence of Brucellosis cases in Neyshabur from 2015–2022.

| Variable Year | Cases | IR[a] | Gender (% of males) | Location (% of rural) | Season (% of patients) | | | |
|---|---|---|---|---|---|---|---|---|
| | | | | | Spring | Summer | Autumn | Winter |
| 2015 | 329 | 64.27 | 60.49 | 83.59 | 38.90 | 35.26 | 13.07 | 12.77 |
| 2016 | 274 | 53.72 | 56.93 | 91.24 | 39.78 | 43.07 | 12.41 | 4.74 |
| 2017 | 292 | 56.95 | 57.19 | 95.21 | 39.04 | 36.30 | 16.44 | 8.22 |
| 2018 | 233 | 46.30 | 54.94 | 93.56 | 31.33 | 40.77 | 13.73 | 14.16 |
| 2019 | 559 | 109.32 | 58.32 | 91.41 | 20.39 | 37.92 | 16.46 | 25.22 |
| 2020 | 640 | 122.21 | 56.56 | 92.81 | 33.91 | 48.75 | 15.47 | 1.88 |
| 2021 | 1356 | 251.94 | 52.95 | 92.12 | 36.95 | 27.58 | 25.52 | 9.96 |
| 2022 | 885 | 161.38 | 53.45 | 79.89 | 32.77 | 28.81 | 17.74 | 20.68 |
| Total | 4568 | 86.79 (54.53–151.59)[b] | 55.36 | 89.32 | 33.84 | 34.76 | 18.63 | 12.76 |

[a] Incidence Rate (IR) per 100,000.

[b] Median (IQR).

According to self-reported data from infected patients during the study period in Neyshabur, 84.60% had direct contact with animals at work or home. Moreover, 64.30% of patients had a history of consuming unpasteurized dairy products, this indicates indirect contact. Among those who had direct or indirect contact with animals, 38.40% were aware that animal vaccination against brucellosis had been performed. Furthermore, 16.70% of patients reported that other family members also were suffered from the disease. See Table 2.

The geographical distribution of disease cases indicates that most infections were concentrated in the southern and southeastern areas of Neyshabur during the study period. See Fig 3.

The results of Moran's Local Indicator of Spatial Association (LISA) test for the districts of Neyshabur reveal that, in six out of the eight years studied (excluding 2016 and 2017), Cold Spot clusters were more frequently found in the northern areas of the city. Conversely,

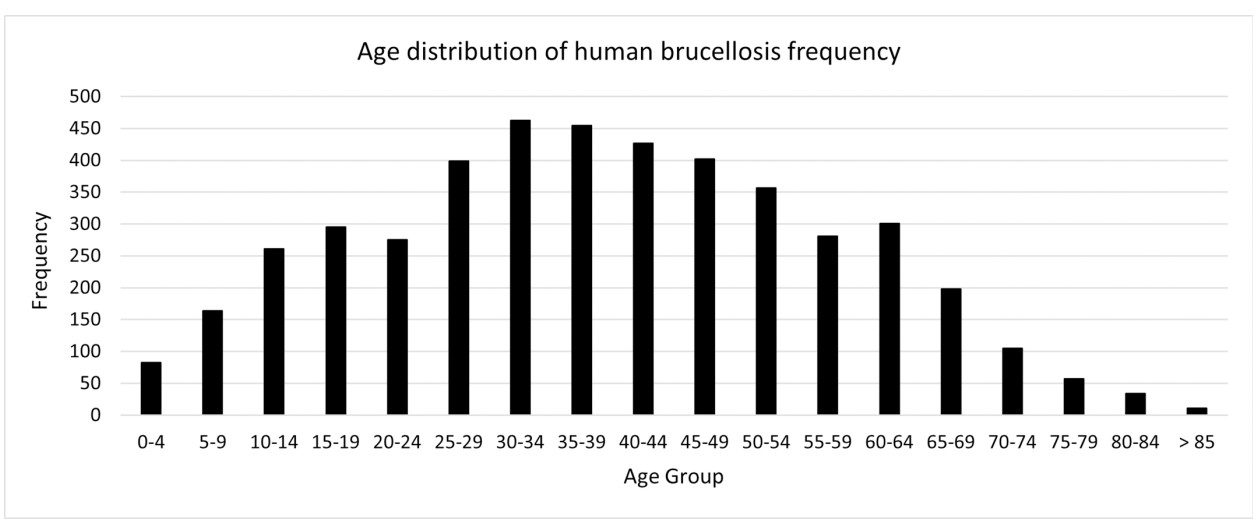

**Fig 1. Age distribution of human brucellosis cases in the Neyshabur, Iran during 2015–2022.**

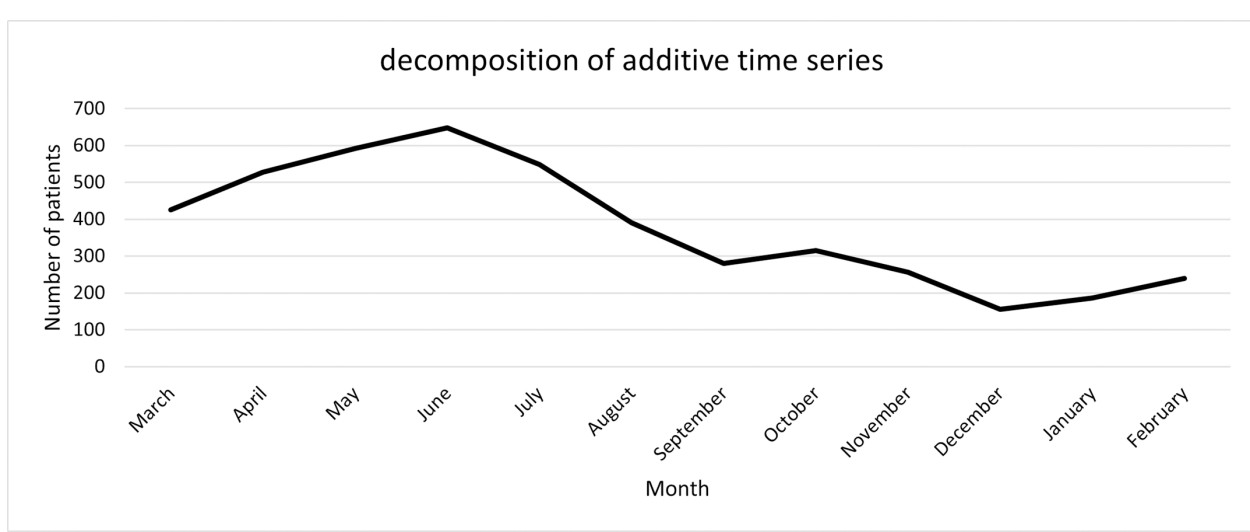

**Fig 2. The monthly distribution of human brucellosis cases in the Neyshabur, Iran during 2015–2022.**

**Table 2.** Distribution of the relative frequencies of the population with brucellosis based on the studied variables.

| Variable Year | Direct interaction with livestock | | | Consuming raw dairy products | | | Livestock vaccination | | | Household member infections | | |
|---|---|---|---|---|---|---|---|---|---|---|---|---|
| | Yes | No | Unclear | Yes | No | Unclear | Yes | No | Unclear | Yes | No | Unclear |
| 2015 | 91.2 | 7.5 | 1.2 | 69.0 | 25.8 | 5.2 | 24.0 | 30.4 | 45.6 | 18.8 | 73.3 | 7.9 |
| 2016 | 90.9 | 6.5 | 2.6 | 66.8 | 25.5 | 7.7 | 18.6 | 12.8 | 68.6 | 20.8 | 69.0 | 10.2 |
| 2017 | 94.5 | 4.5 | 1.0 | 71.2 | 21.3 | 7.5 | 18.5 | 37.3 | 44.2 | 13.7 | 75.3 | 11.0 |
| 2018 | 91.4 | 6.0 | 2.6 | 80.3 | 12.0 | 7.7 | 29.2 | 38.6 | 32.2 | 14.6 | 73.0 | 12.4 |
| 2019 | 90.7 | 8.6 | 0.7 | 69.6 | 28.4 | 2.0 | 37.7 | 49.4 | 12.9 | 17.5 | 73.0 | 9.5 |
| 2020 | 93.3 | 6.7 | 0.0 | 56.6 | 43.4 | 0.0 | 37.8 | 47.4 | 14.8 | 22.0 | 76.6 | 1.4 |
| 2021 | 89.8 | 9.7 | 0.5 | 57.6 | 34.8 | 7.6 | 43.1 | 33.1 | 23.8 | 16.7 | 73.2 | 10.1 |
| 2022 | 57.4 | 38.1 | 4.5 | 67.7 | 28.7 | 3.6 | 52.6 | 38.6 | 8.8 | 7.9 | 85.5 | 6.6 |
| Total | 84.7 | 13.7 | 1.6 | 64.3 | 30.8 | 4.9 | 38.4 | 37.3 | 24.3 | 16.0 | 75.9 | 8.1 |

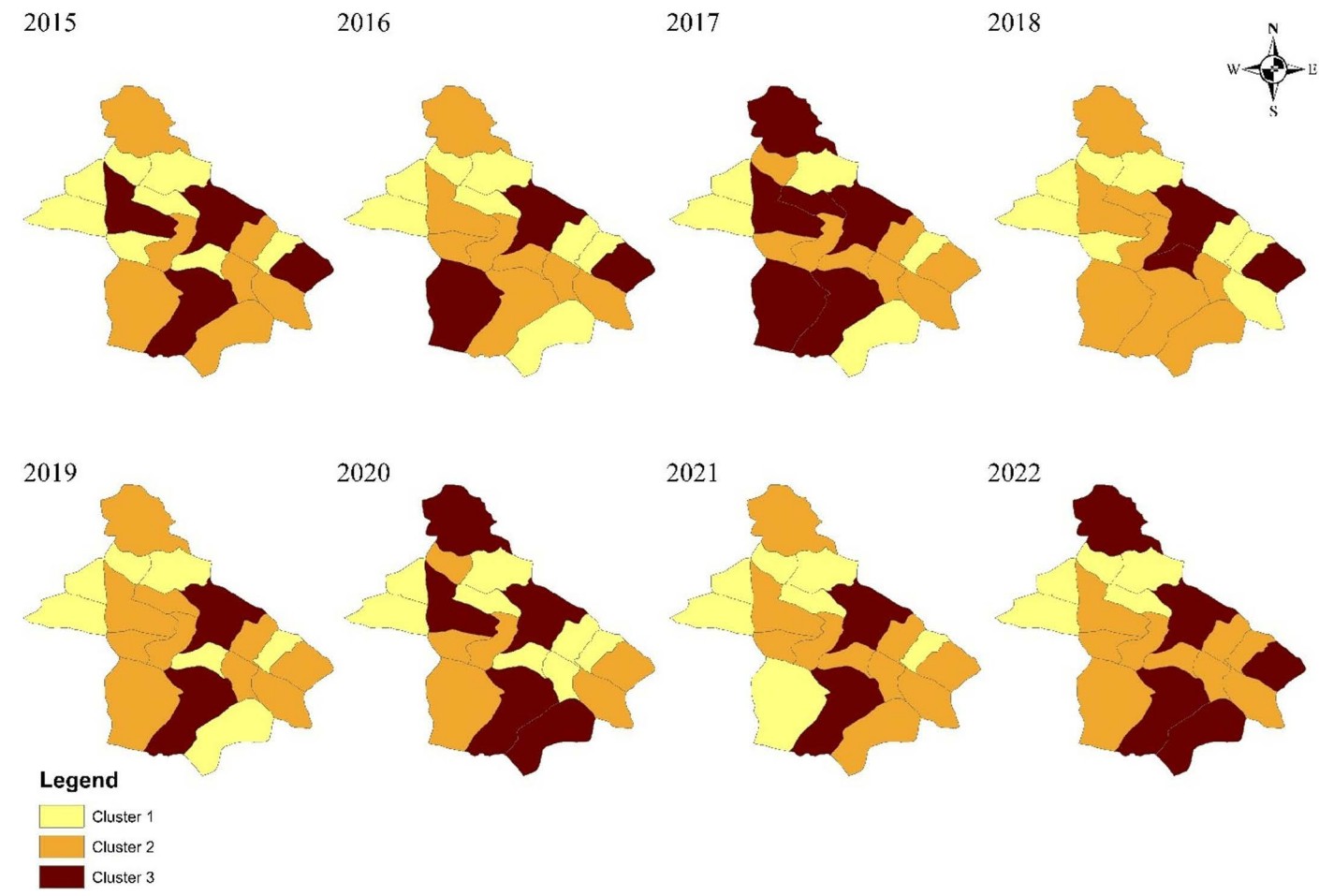

**Fig 3. Geographical distribution of human brucellosis cases at district level in Neyshabur, Iran during 2015–2022.**

Hot Spot clusters were observed in the southwestern region during 2017 and 2019. Additionally, Low-High Outlier clusters were noted in the southeastern region during 2017, 2019, and 2021. See Fig 4.

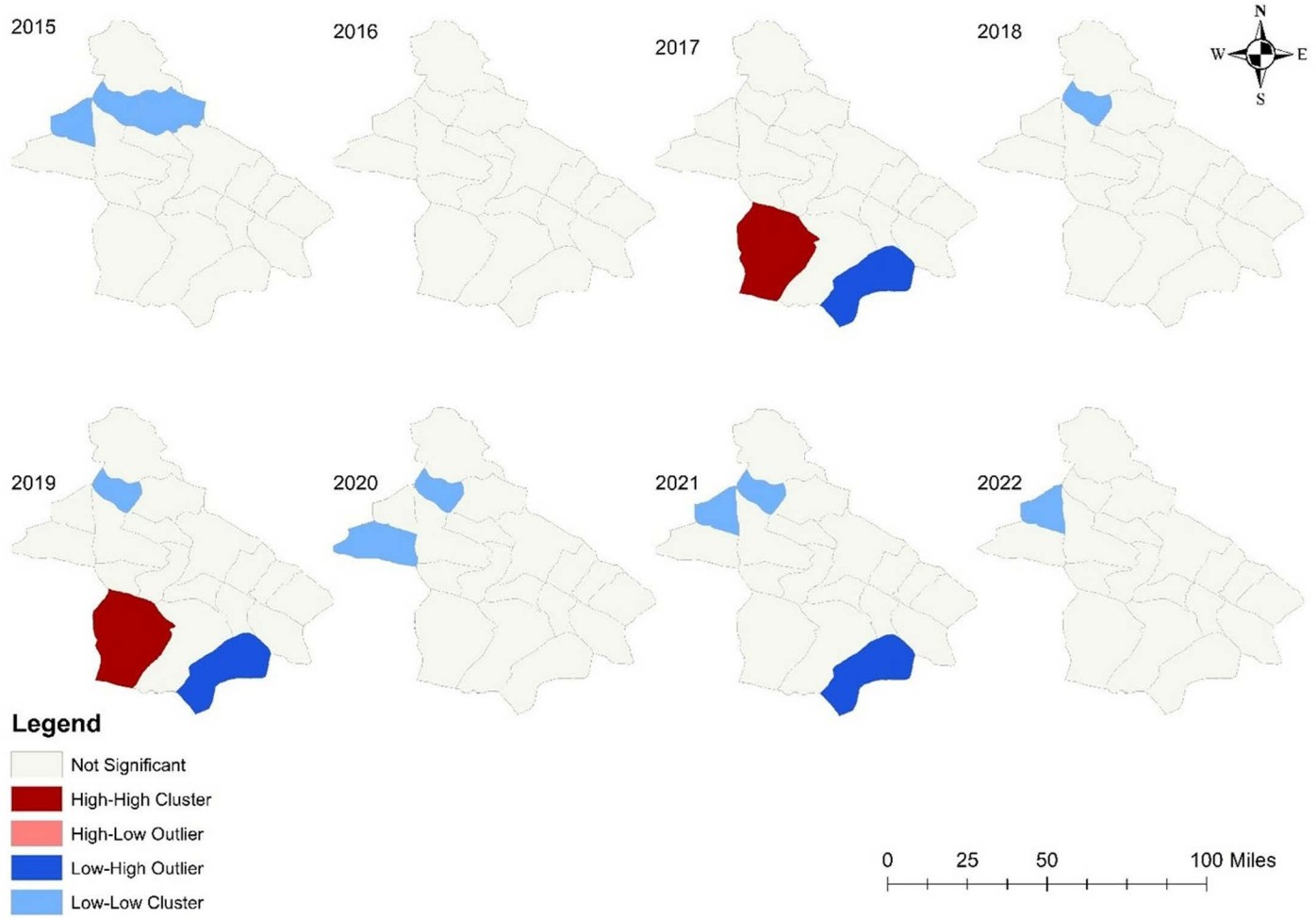

**Fig 4. Spatial cluster analysis of human brucellosis in district level of Neyshabur, Iran during 2015–2022.**

The results of the global Moran's I spatial test, used to determine the geographical distribution of human brucellosis in Neyshabur from 2015 to 2022, showed that the Moran's I index ranged from 0.0123 to 0.2860. A Z-score of less than 1.96 and P-values greater than 0.05 indicate that the geographic distribution of human brucellosis in Neyshabur was random during the study period. See Table 3.

## Discussion

During 2015–2022, a total of 4,568 human brucellosis cases were reported in Neyshabur, with an annual incidence rate 108.26 per 100,000 inhabitants. Although this is likely underestimated as it only includes patients who visited by the governmental health facilities and doesn't account for those who sought treatment at private clinics. In addition, since the general signs and symptoms of brucellosis (e.g., fever, shivering, sweating, weakness, fatigue, headache, lumbar pain and etc.) are vague and similar to influenza, it may lead to delays in the early diagnosis of disease and doing diagnostic tests [36]. Even considering undiagnosed cases, the incidence rate of the disease in Neyshabur is higher than the average for Razavi Khorasan Province. Base on the results conducted by Norouzinezhad, the incidence rate of brucellosis

**Table 3. Global auto-correlation Moran's I value of Human Brucellosis in Neyshabur, Iran during 2015–2022.**

| Year | Moran's I | Variance | Z score | P value | Aggregation |
|------|-----------|----------|---------|---------|-------------|
| 2015 | 0.184453 | 0.048362 | 1.091377 | 0.275107 | No |
| 2016 | 0.110425 | 0.048846 | 0.751000 | 0.452653 | No |
| 2017 | 0.012333 | 0.050060 | 0.303427 | 0.761565 | No |
| 2018 | 0.055398 | 0.047653 | 0.508272 | 0.611263 | No |
| 2019 | 0.240794 | 0.044657 | 1.402355 | 0.160809 | No |
| 2020 | 0.245693 | 0.049658 | 1.351850 | 0.176423 | No |
| 2021 | 0.146019 | 0.048417 | 0.916091 | 0.359619 | No |
| 2022 | 0.286051 | 0.049117 | 1.541381 | 0.123224 | No |

in Razavi Khorasan Province was reported 35.17, 29.52, and 39.30 in 2015, 2016, and 2017, respectively, while during the same years, the incidence rate in Neyshabur was 64.27, 53.72, and 56.95, respectively [11]. Regarding the rising trend in the incidence rate in Neyshabur from 2018 to 2022 indicates a clear need for more detailed planning to control the disease in the coming years.

Our findings showed that the ratio of rural to urban residents affected by brucellosis is approximately 9 to 1. Consistent with previous studies, the rural population is at a higher risk of contracting the disease [11,25,37,38]. This difference in infection rates may be associated with the rural lifestyle in Iran, where most people are engaged in agriculture and livestock farming. Nevertheless, a study conducted in Semnan Province found that 70.9% of patients lived in urban areas [39]. Some internal and external studies have identified brucellosis as an occupational disease [11,31,40,41]. This finding supports our results, which indicates no significant difference in brucellosis incidence between genders, with a male-to-female ratio of 1.24. Most similar Iranian study also report no meaningful difference between males and females [42,43] this may be attributed to the fact that, in rural areas of Iran, women are similarly involved in animal farming as men, taking on responsibilities such as cleaning pens and milking [44]. Although some studies in China indicate a significant difference in the sex ratio, with values of 2.64 [40] and 2.5 [45] (male/female), these variations may be attributed to differences in lifestyle across societies. The average age of men affected by brucellosis is 35.68 years, compared to 42.37 years for women. This notable difference may be due to men entering livestock farming at a younger age.

Most studies have shown direct contact with livestock and consumption of raw dairy products as risk factors for brucellosis. Consistent with our findings, several studies have also highlighted direct contact with livestock as the most prevalent risk factor for the brucellosis in humans [12,25]. Almost 85% of patients reported that they are in direct contact with animals at their workplace or living place. Given that 90% of rural residents affected by brucellosis contacting to animals, this exposure can be seen as a key risk factor. Additionally, our findings reveal that about 65% of patients had a history of consuming raw dairy products. Previous studies reported raw dairy consumption as the most common risk factor for brucellosis, which differs from our results [18,42,46] In rural areas, there is a common belief that unprocessed dairy products are more nutritious than pasteurized ones. This perception increases the preference for consuming raw dairy products, such as milk, yogurt, and cream [11]. Educating programs, particularly among women who are more involved in family nutrition, could enhance awareness and reduce the consumption of raw dairy products. Our findings indicate peak incidence between March and June, with a subsequent decrease until reaching its lowest level in December. The infection rate subsequently rises again, reaching its peak in June. Similar studies conducted in various provinces of Iran [37,38,18,43], and in Shenyang, China [47]

reported the highest incidence of brucellosis in June and July, indicating a summer peak. In contrast, studies in Bosnia and Fars, Iran, reported peak of incidence in spring [48] and winter [31] respectively. Moreover, the seasonal increase during summer is likely associated with the breeding season. During this period, contact with fetal secretions during and after childbirth, combined with the increased production and consumption of unpasteurized milk and dairy products in summer, contributes to a higher incidence of brucellosis [11,42,49]. To mitigate this, educational programs focusing on protective measures against contact with livestock and their secretions, as well as reducing the consumption of raw dairy products during the summer months, could be effective in preventing brucellosis.

The spatial clustering analysis of brucellosis incidence across 19 districts in Neyshabur from 2015 to 2022 revealed spatial clusters in all years except 2016. The distribution of these clusters was consistent over the years, with high incidence rates often observed in the southwestern part of the city. This region is characterized by poorer and more isolated villages, which often lack adequate health infrastructure and monitoring, particularly considering livestock exchange between neighboring villages. This shows insufficient attention to livestock immunization, inadequate quarantine actions in villages with high brucellosis incidence, and a lack of effective monitoring of public health in these areas. Additionally, clusters with low incidence have consistently been observed in the northern and northwestern regions over the years. The lower incidence of brucellosis in these areas is likely due to the low population density in the villages. This region, situated between the cities of Quchan and Khoshab in Razavi Khorasan Province, is known for its small population compared to other villages. Between 2017 and 2019, as well as in 2021, the Belherat district, located near high-incidence areas, was characterized by a low incidence of the disease. A limitation of our study is the inability to determine the reasons for this spatial variation in the southeastern part of Neyshabur during these years.

The results of Moran's spatial Autocorrelation (Global Moran's I) indicate that over the eight years studied, the distribution of human brucellosis in Neyshabur's villages was random, with no consistent pattern of high or low incidence. This suggests that there is a need for more comprehensive and precise studies to better predict brucellosis incidence in Neyshabur City, especially given the rising trend of the disease. Enhanced and detailed research is necessary to address the increasing rates of brucellosis in this city.

## Conclusion

The findings of this study can inform the planning of educational initiatives, particularly those targeting mothers, as well as strategies for animal vaccination. These efforts should be emphasized during the months leading up to the peak human brucellosis season and in regions with high incidence rates. Future research should explore the extent of livestock vaccination coverage and its correlation with the incidence of human brucellosis.

### Limitations & strength

Due to inadequate accurate registration of the livestock numbers by districts, the lack of access to local animal vaccination data based on the geographical distribution of livestock farming, and the deficiency in measuring the annual rainfall at the district level in Neyshabur, the present study didn't examine the effect of these factors on the annual incidence rate of human brucellosis. Despite these limitations, we indicated the spatial pattern of human brucellosis incidence in Neyshabur and increased our knowledge about the most common causes of the disease in the human population. This information can be used as a scientific resource for revising prevalence strategies to control or eradicate human brucellosis in Neyshabur.

## Suggestions and applications

It is recommended that the results of this study be utilized to develop a plan aimed at preventing the increase in the incidence of human brucellosis in Neyshabur and other cities and provinces across the country. For future studies, it is advisable to focus on the effects of additional risk factors, such as rainfall amounts, crop cover in the area, and the impact of livestock vaccination on human brucellosis.

## Acknowledgments

This study was conducted under the ethics code IR.NUMS.REC.1401.042. the authors would like to extend their sincere gratitude to Dr. Benyamin Hosseini for his valuable comments and suggestions that allowed us to improve the quality of the study. Additionally, we are especially grateful for the help of Dr. Ensiyeh Mollanoroozy for her expert editing support, who spent hours to strengthen the overall quality of the manuscript beyond correcting grammatical and spelling mistakes.

## Author contributions

**Data curation:** Fatemeh Khorashadizadeh.

**Formal analysis:** Fatemeh Khorashadizadeh.

**Funding acquisition:** Mohammad Maroosi.

**Investigation:** Ayoub Fathabadi, Reza Darrudi.

**Methodology:** Reza Darrudi.

**Project administration:** Mohammad Maroosi.

**Supervision:** Mohammad Maroosi.

**Validation:** Fatemeh Khorashadizadeh.

**Visualization:** Ayoub Fathabadi.

**Writing – original draft:** Ayoub Fathabadi.

**Writing – review & editing:** Ayoub Fathabadi.

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
