## [Decision Letter · Decision Letter 0]

18 Jun 2024

PONE-D-24-16267Spatio-temporal analysis of human brucellosis distribution in Neyshabur, Iran from 2015 to 2022PLOS ONE

Dear Dr. Maroosi,

Thank you for submitting your manuscript to PLOS ONE. After careful consideration, we feel that it has merit but does not fully meet PLOS ONE’s publication criteria as it currently stands. Therefore, we invite you to submit a revised version of the manuscript that addresses the points raised during the review process.

We look forward to receiving your revised manuscript.

Kind regards,

Maryam Afshari

Academic Editor

PLOS ONE

Journal Requirements:

2. Thank you for stating the following financial disclosure: "This study was supported by the Student Research Committee (SRC) funds of Neyshabur University of Medical Sciences (NUMS)." 

4. Please include your figure as part of your main manuscript and remove the individual files. Please note that supplementary figure (should remain/ be uploaded) as separate ""supporting information"" files"

Reviewers' comments:

Reviewer's Responses to Questions

**Comments to the Author**

1. Is the manuscript technically sound, and do the data support the conclusions?

Reviewer #1: Partly

Reviewer #2: Partly

2. Has the statistical analysis been performed appropriately and rigorously? 

Reviewer #1: Yes

Reviewer #2: N/A

3. Have the authors made all data underlying the findings in their manuscript fully available?

Reviewer #1: No

Reviewer #2: Yes

4. Is the manuscript presented in an intelligible fashion and written in standard English?

Reviewer #1: Yes

Reviewer #2: Yes

5. Review Comments to the Author

Reviewer #1: 1. In the introduction, researchers should justify the necessity of spatio-temporal analysis of brucellosis in Neyshabur. Neyshabur is a small city in the east of Iran. The main centers of brucellosis are concentrated in the western provinces of Iran. What are the spatial units in Neishabur city?

2. Due to the fact that the result was incomplete and I did not have access to the figures, I cannot fully review the article.

Reviewer #2: - You can change the title to news and mention the type of study in the title

- You can write Neyshabur, a city in the east of Iran

- The working method of the study is written in the abstract in a very limited way. The exact method of implementation should be written

- In the conclusion section for this study, it should be written and applied

- Match keywords with mesh

- In the introduction, it should be explained how most brucellosis is transmitted and how it is in Iran

- Where are the most involved areas in Iran?

- Write the occurrence and prevalence by mentioning the figures

- There was no explanation given in the introduction about what was the reason to do this study in Neishabur

- Most of the information and statistics are for 2017, you can update it until 2023

- Write down other studies that exist in this field especially in the east of Iran

- Write the criteria for entering the study

- Rewrite the study implementation method section in the work method, it is written very vaguely and scattered

- In the results section, you have only brought a table, while you can bring some findings in the form of a table

- The discussion of the study is also written very briefly

- In the discussion after your findings, the author's own analysis should be written in this context

- Bring studies both similar and contradictory to these results

- In the discussion part, the results of all the cities in the field of Brucellosis, especially the province in question and the neighboring cities, should be used

- The conclusion is too long

- Focus on the result of this study and its applications

- Mention the suggestions in this context

- More for the results of your own study, you should continue drawing conclusions

6. PLOS authors have the option to publish the peer review history of their article (what does this mean? ). If published, this will include your full peer review and any attached files.

**Do you want your identity to be public for this peer review?** For information about this choice, including consent withdrawal, please see our Privacy Policy .

Reviewer #1: No

Reviewer #2: No

---

## [Author Response · Author response to Decision Letter 0]

9 Sep 2024

Dear Editor,

Thank you for your feedback on our submission. We understand the importance of data sharing and transparency in research. However, as we previously mentioned, the data used in our study was obtained from the Neyshabur Health Center in the form of an offline document or Excel file. Unfortunately, we are not the owners of this data and cannot provide a direct link for access.

In addition, the Neyshabur Health Center is a local service-oriented institution without the infrastructure for handling international correspondence, especially in languages other than Persian. There is no established protocol or mechanism in place for responding to data requests in foreign languages or from international researchers. This limitation makes it infeasible to provide any direct contact information for the Health Center regarding data access.

That said, while we are unable to make the data publicly available, we are willing to share the data with individual researchers upon request, provided that the request aligns with the ethical guidelines and constraints under which the data was collected.

furthermore, the cleaned dataset contain geographial coordinate of patients and it can be considered as sensitive data and restrict data sharing

Given these constraints, we respectfully request that the data sharing requirements be reconsidered in the context of the limitations we face. We assure you that all data used in this study was handled in accordance with local ethical guidelines, and any data sharing beyond our current capabilities is unfortunately not possible.

We appreciate your understanding and look forward to your guidance on how to proceed under these circumstances.

Best regards,

Mohammad Maroosi

---

## [Decision Letter · Decision Letter 1]

29 Sep 2024

PONE-D-24-16267R1Spatio-temporal analysis of human brucellosis distribution in Neyshabur, Iran from 2015 to 2022, a cross-sectional studyPLOS ONE

Dear Dr. Maroosi,

Thank you for submitting your manuscript to PLOS ONE. After careful consideration, we feel that it has merit but does not fully meet PLOS ONE’s publication criteria as it currently stands. Therefore, we invite you to submit a revised version of the manuscript that addresses the points raised during the review process.

We look forward to receiving your revised manuscript.

Kind regards,

Maryam Afshari

Academic Editor

PLOS ONE

Journal Requirements:

Additional Editor Comments:

Hello

One of the referees has explained in the text that the article was accepted but rejected and according to his opinion the article is acceptable and the next referee asked for corrections in the second revision, which the respected author should answer again.

thanks

Reviewers' comments:

Reviewer's Responses to Questions

**Comments to the Author**

1. If the authors have adequately addressed your comments raised in a previous round of review and you feel that this manuscript is now acceptable for publication, you may indicate that here to bypass the “Comments to the Author” section, enter your conflict of interest statement in the “Confidential to Editor” section, and submit your "Accept" recommendation.

Reviewer #1: All comments have been addressed

Reviewer #2: (No Response)

2. Is the manuscript technically sound, and do the data support the conclusions?

Reviewer #1: Partly

Reviewer #2: Partly

3. Has the statistical analysis been performed appropriately and rigorously? 

Reviewer #1: Yes

Reviewer #2: Yes

4. Have the authors made all data underlying the findings in their manuscript fully available?

Reviewer #1: No

Reviewer #2: Yes

5. Is the manuscript presented in an intelligible fashion and written in standard English?

Reviewer #1: Yes

Reviewer #2: Yes

6. Review Comments to the Author

Reviewer #1: Dear author(s)

I hope this message finds you well. I am writing to inform you that your article entitled “Spatio-temporal analysis of human brucellosis distribution in Neyshabur, Iran from 2015 to 2022, a cross-sectional study” (Manuscript ID: PONE-D-24-16267R1) is acceptable in current form and can be publish in Plos One.

Sincerely, Fatemeh Shahbazi

Assistant Professor of Epidemiology Department of Epidemiology School of Public Health Hamadan University of Medical Sciences, Hamadan, Iran.

Shahbazif2017@gmail.com

Reviewer #2: - You should not write the code of ethics in the abstract part of the work procedure

You should put it in the appropriate place at the end of the conclusion

- - In the results section of the abstract, use the statistics of the results section

- Match the key blocks with the mesh

- Complete the texts in the introduction section and use more relevant studies in this field

- In the introduction section, write the importance and reasons for conducting the study

- The text of the article needs English editing and writing, especially in the results and discussion and working methods

- More internal and external studies should be used in the discussion

- Use both similar and contradictory results

- Write the limitations of the study

- Write suggestions and applications

7. PLOS authors have the option to publish the peer review history of their article (what does this mean? ). If published, this will include your full peer review and any attached files.

**Do you want your identity to be public for this peer review?** For information about this choice, including consent withdrawal, please see our Privacy Policy .

Reviewer #1: **Yes: ** Dr. Fatemeh Shahbazi

Reviewer #2: No

---

## [Author Response · Author response to Decision Letter 1]

7 Nov 2024

Dear Editors, we upload figures to the Preflight Analysis and Conversion Engine (PACE) based on your comments and according to the structure of the journal article, the figures were removed from the manuscript and attached in a compressed file labeled with the names of the figures.

---

## [Decision Letter · Decision Letter 2]

3 Jan 2025

PONE-D-24-16267R2Spatio-temporal analysis of human brucellosis distribution in Neyshabur, Iran from 2015 to 2022, a cross-sectional studyPLOS ONE

Dear Dr. Maroosi,

Thank you for submitting your manuscript to PLOS ONE. After careful consideration, we feel that it has merit but does not fully meet PLOS ONE’s publication criteria as it currently stands. Therefore, we invite you to submit a revised version of the manuscript that addresses the points raised during the review process.

We look forward to receiving your revised manuscript.

Kind regards,

Maryam Afshari

Academic Editor

PLOS ONE

Journal Requirements:

Reviewers' comments:

Reviewer's Responses to Questions

**Comments to the Author**

1. If the authors have adequately addressed your comments raised in a previous round of review and you feel that this manuscript is now acceptable for publication, you may indicate that here to bypass the “Comments to the Author” section, enter your conflict of interest statement in the “Confidential to Editor” section, and submit your "Accept" recommendation.

Reviewer #1: All comments have been addressed

Reviewer #3: (No Response)

2. Is the manuscript technically sound, and do the data support the conclusions?

Reviewer #1: Partly

Reviewer #3: Yes

3. Has the statistical analysis been performed appropriately and rigorously? 

Reviewer #1: I Don't Know

Reviewer #3: Yes

4. Have the authors made all data underlying the findings in their manuscript fully available?

Reviewer #1: Yes

Reviewer #3: No

5. Is the manuscript presented in an intelligible fashion and written in standard English?

Reviewer #1: Yes

Reviewer #3: Yes

6. Review Comments to the Author

Reviewer #1: (No Response)

Reviewer #3: Have the authors made all data underlying the findings in their manuscript fully available? No

The authors have declared some level of restrictions to data availability.

7. PLOS authors have the option to publish the peer review history of their article (what does this mean? ). If published, this will include your full peer review and any attached files.

**Do you want your identity to be public for this peer review?** For information about this choice, including consent withdrawal, please see our Privacy Policy .

Reviewer #1: **Yes: ** Fatemeh Shahbazi

Reviewer #3: No

---

## [Author Response · Author response to Decision Letter 2]

8 Jan 2025

Thank you for your inquiry regarding data sharing. We appreciate the importance of transparency and accessibility in research.

The data used in this study was obtained from the Neyshabur Health Center as offline documents and Excel files. As the authors do not own this data, we are unable to provide a direct link or public access. Additionally, the Neyshabur Health Center is a local service-oriented institution that lacks the infrastructure for handling international correspondence, particularly in languages other than Persian. There is currently no established protocol for responding to data requests from international researchers, which limits our ability to facilitate direct contact for data access.

Moreover, the data includes sensitive information, such as geographical coordinates of patients, imposing further restrictions on sharing due to privacy and ethical considerations.

Despite these constraints, we are committed to sharing the data with individual researchers upon request, provided that such requests comply with the ethical guidelines and constraints under which the data was collected. The data will be stored securely in a personal and online encrypted cloud repository to ensure long-term availability.

While we cannot make our study’s data publicly available at the time of publication, we assure you that all authors are committed to making the data underlying the findings described in this study fully available without restriction to those who request it, in compliance with the PLOS Data Availability policy. For datasets involving personally identifiable information or other sensitive data, data sharing will be contingent on the data being handled appropriately by the requester and in accordance with all applicable local requirements.

Thank you for your understanding.

---

## [Decision Letter · Decision Letter 3]

17 Jan 2025

Spatio-temporal analysis of human brucellosis distribution in Neyshabur, Iran from 2015 to 2022, a cross-sectional study

PONE-D-24-16267R3

Dear Dr. Maroosi,

We’re pleased to inform you that your manuscript has been judged scientifically suitable for publication and will be formally accepted for publication once it meets all outstanding technical requirements.

Kind regards,

Maryam Afshari

Academic Editor

PLOS ONE

Additional Editor Comments (optional):

Reviewers' comments:

Reviewer's Responses to Questions

**Comments to the Author**

1. If the authors have adequately addressed your comments raised in a previous round of review and you feel that this manuscript is now acceptable for publication, you may indicate that here to bypass the “Comments to the Author” section, enter your conflict of interest statement in the “Confidential to Editor” section, and submit your "Accept" recommendation.

Reviewer #3: All comments have been addressed

2. Is the manuscript technically sound, and do the data support the conclusions?

Reviewer #3: Yes

3. Has the statistical analysis been performed appropriately and rigorously? 

Reviewer #3: Yes

4. Have the authors made all data underlying the findings in their manuscript fully available?

Reviewer #3: Yes

5. Is the manuscript presented in an intelligible fashion and written in standard English?

Reviewer #3: Yes

6. Review Comments to the Author

Reviewer #3: (No Response)

7. PLOS authors have the option to publish the peer review history of their article (what does this mean? ). If published, this will include your full peer review and any attached files.

**Do you want your identity to be public for this peer review?** For information about this choice, including consent withdrawal, please see our Privacy Policy .

Reviewer #3: No

---

## [Editor Report · Acceptance letter]

PONE-D-24-16267R3

PLOS ONE

Dear Dr. Maroosi,

I'm pleased to inform you that your manuscript has been deemed suitable for publication in PLOS ONE. Congratulations! Your manuscript is now being handed over to our production team.

Kind regards,

on behalf of

Dr. Maryam Afshari

Academic Editor

PLOS ONE